# R2TP/Prefoldin-like component RUVBL1/RUVBL2 directly interacts with ZNHIT2 to regulate assembly of U5 small nuclear ribonucleoprotein

Philippe Cloutier[1], Christian Poitras[1], Mathieu Durand[2], Omid Hekmat[1], Émilie Fiola-Masson[1], Annie Bouchard[1], Denis Faubert[1], Benoit Chabot[2,3] & Benoit Coulombe[1,4]

The R2TP/Prefoldin-like (R2TP/PFDL) complex has emerged as a cochaperone complex involved in the assembly of a number of critical protein complexes including snoRNPs, nuclear RNA polymerases and PIKK-containing complexes. Here we report on the use of multiple target affinity purification coupled to mass spectrometry to identify two additional complexes that interact with R2TP/PFDL: the TSC1–TSC2 complex and the U5 small nuclear ribonucleoprotein (snRNP). The interaction between R2TP/PFDL and the U5 snRNP is mostly mediated by the previously uncharacterized factor ZNHIT2. A more general function for the zinc-finger HIT domain in binding RUVBL2 is exposed. Disruption of ZNHIT2 and RUVBL2 expression impacts the protein composition of the U5 snRNP suggesting a function for these proteins in promoting the assembly of the ribonucleoprotein. A possible implication of R2TP/PFDL as a major effector of stress-, energy- and nutrient-sensing pathways that regulate anabolic processes through the regulation of its chaperoning activity is discussed.

[1] Translational Proteomics Laboratory, Institut de Recherches Cliniques de Montréal (IRCM), Montreal, Quebec, Canada H2W 1R7. [2] Laboratory of Functional Genomics, Faculté de Médecine et des Sciences de la Santé, Université de Sherbrooke, Sherbrooke, Quebec, Canada J1E 4K8. [3] Département de Microbiologie et d'Infectiologie, Faculté de Médecine et des Sciences de la Santé, Université de Sherbrooke, Sherbrooke, Quebec, Canada J1E 4K8. [4] Département de Biochimie, Université de Montréal, Montreal, Quebec, Canada H3T 1J4. Correspondence and requests for materials should be addressed to B.C. (email: benoit.coulombe@ircm.qc.ca).

Protein chaperones are factors that assist in the folding of newly synthesized 'client' polypeptides, ensure their integration within larger molecular complexes, prevent or resolve their aggregation, modulate their activity by maintaining otherwise unstable conformation and/or facilitate switching between multiple functional conformational states. Chaperones often require non-client proteins, termed 'cochaperones', to favour steps of the nucleotide hydrolysis cycle upon which most chaperones operate. Cochaperones also help drive specificity and physically link chaperones together or with other molecular machineries that can ultimately impact on the modification, localization and turnover of client proteins.

The recently discovered R2TP/Prefoldin-like (R2TP/PFDL) complex is unique among chaperone cofactors in that it provides a platform upon which an unparalleled number of different chaperones gather. Firstly, the dual tetratricopeptide repeat (TPR) domains of RPAP3, a subunit of R2TP/PFDL, can bind to both Hsp70 and Hsp90 (refs 1,2) in a manner akin to its closest paralog, STIP1/Hop. Also, and as the name implies, this complex contains a prefoldin-like module. The canonical prefoldin complex (PFD) is best known for assisting in the folding of nascent cytoskeletal proteins actin and α and β tubulin with the help of the Chaperonin containing TCP-1 (CCT) complex[3,4]. While two subunits (PFDN2, PFDN6) are shared between PFD and the prefoldin-like module, three are specific to the latter (URI1, PDRG1, UXT). Finally, R2TP/PFDL also comprises the AAA+ ATPases RUVBL1 and RUVBL2. Many AAA+ ATPases are well-known protein chaperones like members of the ClpB/Hsp104 family, conserved from prokaryotes to eukaryotes, whose unfoldase activity assists in protein disaggregation[5]. While not all AAA+ ATPases are chaperones in the strictest sense of the term, many of them do display modes of action that are mechanistically comparable to that of typical chaperone to fulfil their function in diverse biological processes as protein degradation, translocation, membrane fusion, trafficking, microtubule severing and DNA replication[6]. Although RUVBL1 and RUVBL2 have not been formally classified as chaperones, the presence of these ATPases in various chromatin remodelling complexes[7], combined with their role in biogenesis of some ribonucleoprotein particles (RNP)[8–12], tend to suggest a similar mode of action in the assembly of protein–nucleic acid complexes.

Involvement of R2TP in box C/D snoRNP biogenesis was the first recognized role of R2TP in *Saccharomyces cerevisiae* (Fig. 1)[10]. The composition of the complex in yeast, which is limited to the two RUVBL proteins (Rvb1 and Rvb2), Tah1 (ortholog of human RPAP3) and Pih1, gave the complex its acronym 'R2TP' (ref. 1). It was subsequently shown that the function of the human equivalent of R2TP was not restricted to box C/D snoRNPs but also included any RNPs harbouring RNA-binding proteins of the L7Ae-family: NHP2L1/15.5K/Snu13 (box C/D snoRNPs, U4 small nuclear ribonucleoprotein (snRNP)), NHP2 (box H/ACA snoRNPs, telomerase) and SECISBP2/SBP2 (selenoprotein mRNAs)[13]. Furthermore it was noted that box C/D assembly required additional factors like NUFIP1, ZNHIT6/BCD1 (ref. 9) and ZNHIT3 (refs 14,15).

The study of soluble human RNA polymerase II (RNAPII) protein interactions led to the discovery of a number of previously uncharacterized RNAPII-associated proteins (RPAPs)[16,17], including R2TP/PFDL (refs 18,19). It was later shown that the complex and Hsp90 chaperone are involved in RNAPII assembly through interaction with an RPB1–RPB8 subcomplex[20]. This mechanism is most likely not limited to RNAPII as prefoldin-like protein URI1 and its *S. cerevisiae* homologue, Bud27p, were shown to also have a function in the biogenesis of RNAPI and III (ref. 21).

More recently, R2TP/PFDL was found in association with Phosphatidylinositol 3-kinase-related kinases (PIKKs)[22]. PIKKs consist of a family of kinases and kinase-related proteins that act in various aspects of cell biology like DNA damage response (ATM, ATR and DNA-PK), integration of nutrient and growth factors signalling (mTOR), non-sense mediated mRNA decay (SMG-1) and chromatin remodelling (TRRAP). Although a complex consisting of TELO2/Tel2, TTI1 and TTI2 (referred to as the TTT complex) has already been implicated in the stability and biogenesis of PIKKs[23–25], it was ultimately discovered that this effect also required the action of Hsp90 and R2TP/PFDL[22,26,27].

The present article reports two additional interacting complexes for R2TP/PFDL: the U5 snRNP, a central component of the spliceosome, and the TSC1–TSC2 complex, an inhibitor of mTOR whose activity is targeted by various signalling pathways in response to growth factors and nutrient availability. Reciprocal purification of both U5 snRNP and TSC1–TSC2 components using selected tagged subunits confirmed association with R2TP/PFDL and revealed many interactors. One of these, the zinc-finger protein ZNHIT2 is shown here to act as a mediator of U5 snRNP interaction with R2TP/PFDL. Furthermore, we identify the general function of the zinc finger HIT domain, present in ZNHIT2 and five other human ZNHIT family members, as a RUVBL2-binding domain. Finally, we show that disruption of ZNHIT2 and RUVBL2 expression levels affect U5 snRNP protein composition, suggesting a role for R2TP/PFDL in assembly of this RNP particle.

## Results

**Affinity purification of R2TP/PFDL reveals interactors.** Results of gel-based tandem affinity purification coupled to mass spectrometry (TAP-MS) of R2TP/PFDL subunits have been reported previously by our group[18,19] and although this dataset did identify interactions with all three RNA polymerases, no evidence was obtained that corroborated the connection to L7Ae RNPs or PIKKs. Over the past few years, advances in mass spectrometry technologies, modifications to our TAP-MS protocol[28] and improvements to our computational analysis of protein–protein interactions[28,29] have made it possible to study purified complexes in solution from various cell fractions with limited starting material. Such improvements have led us to re-examine the protein interaction network of the R2TP/PFDL complex.

As a result, we are now able to identify with high-confidence box C/D snoRNP component NOP58, as well as box H/ACA snoRNP subunits SHQ1 and DKC1 (refs 30–32), thereby confirming association with at least a subset of L7Ae RNPs and suggesting that it is indeed involved in their biogenesis (Fig. 2a, Supplementary Data 1). The only detected PIKK was TRRAP in the chromatin fraction of the RUVBL2 purification, although mTORC2 subunit RICTOR and SMG-1 interactor UPF1 were identified. Notably, all three subunits of the TTT complex (TELO2, TTI1 and TTI2) did copurify with R2TP/PFDL subunits. As was the case in our initial experiments[18,19], protein components of all three RNAPs are among the strongest interactors of the R2TP/PFDL complex, as were a number of associated factors including RPAP2, SLC7A6OS and TANGO6, which have been shown to be involved in RNAP biogenesis and nuclear import[33–35].

Among identified interactors of the R2TP/PFDL complex were several protein subunits of the U5 snRNP (AAR2, EFTUD2, PRPF8 and SNRNP200) and the TSC1–TSC2 complex (TSC1, TSC2 and TBC1D7). To confirm these interaction data we gathered from TAP-MS and to address whether these

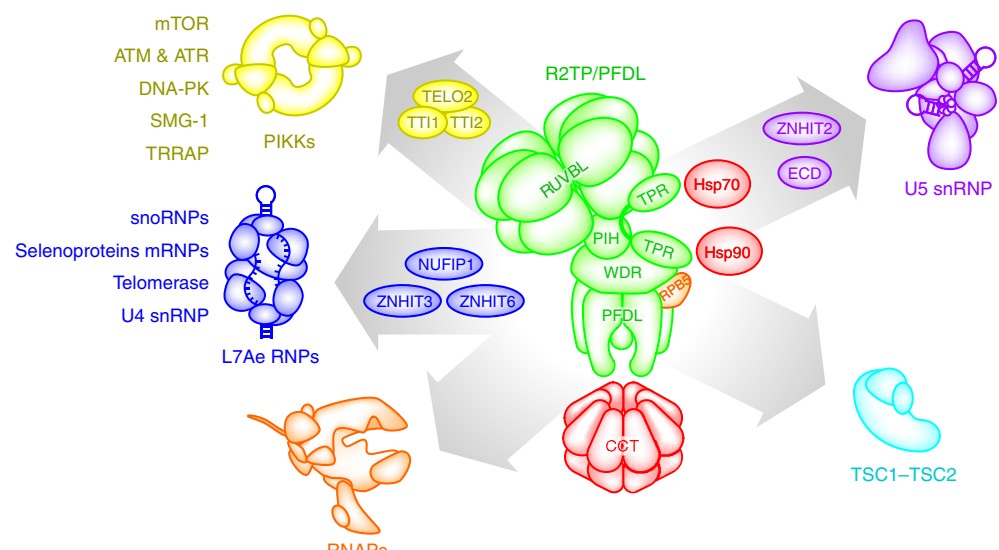

**Figure 1 | Interactors of the R2TP/PFDL chaperone.** Representation of known interactions (left) and previously undescribed (right) interactions of the R2TP/PFDL complex described in this study. Associated chaperones are shown in red and known or putative cofactors for each client complex are identified inside arrows.

interactions are cell-specific, CoImmunoPrecipitation (CoIP) of FLAG-tagged subunits of the R2TP/PFDL complex were performed in HeLa cells (Fig. 2b). Through western blotting, endogenous TSC1–TSC2 complex subunits TSC1 and TSC2 as well as U5 snRNP components PRPF8/Prp8 and EFTUD2/Snu114 were all shown to associate with ectopically expressed FLAG-URI1 and FLAG-RPAP3 with relatively similar efficiency.

The U5 snRNP is one of five major snRNP that make up the major spliceosome that catalyses removal of introns from newly transcribed pre-mRNAs (Fig. 2c). Essentially, the stepwise splicing reaction begins with recognition of the 5′ end of the intron, also known as 5′ splice site, by U1 snRNP as well as elements in and around the 3′ splice site by U2 snRNP and its auxiliary factor (U2AF). A larger complex made up of U4, U5 and U6 snRNPs termed 'U4/U6.U5 tri-snRNP' makes contact with initial recognition modules and thus brings both ends of the intron in close proximity. Significant helicase-driven rearrangements in the secondary structure of snRNAs lead to hybridization of U2 and U6 snRNAs and release of U4 and U1 snRNPs from the spliceosome. During this time, U5 snRNP stabilizes association with 5′ and 3′ splice sites through RNA/RNA and protein/RNA interactions. Considerable reshuffling is also noted in the overall protein content of the spliceosome as the NineTeen Complex (NTC, named for central component PRPF19/Prp19) integrates with the spliceosome and remains tightly coupled to U5 snRNP[36]. Following this reorganization, the spliceosome is finally catalytically active and can proceed with excision of the intron and splicing of adjoining exons by two sequential transesterification reactions. Interestingly, there is a second, minor spliceosome that carries out excision of a subset of introns (representing <1% of total introns encoded in the human genome) and although U1, U2, U4 and U6 snRNPs all have counterparts that are specialized for this particular spliceosome (denoted U11, U12, U4atac and U6atac), U5 is the only snRNP that is common to both[37].

TSC1 and TSC2 are tumour suppressors whose genes have been found to be mutated in the multisystemic tumour syndrome, tuberous sclerosis. TSC2 possesses a GTPase activating protein (GAP) activity directed towards G-protein Rheb (Fig. 2d). Upon hydrolysis of GTP, GDP-bound Rheb releases the PIKK

complex mTORC1, leaving it inactive[38]. This ultimately leads to hindrance of cell growth and proliferation by shutting down anabolic processes. Under stress conditions like hypoxia and low energy or in response to growth factors and cytokines, activity of TSC1–TSC2 can be either up- or downregulated through phosphorylation of different sites on both components by specific kinases. More recently, an additional subunit of the TSC1–TSC2 complex has been identified, TBC1D7/TBC7, which also shares homology with GAPs although it is still unclear which G-protein, if any, it might regulate[39,40].

**R2TP/PFDL is a major interactor of the TSC1–TSC2 complex.** Having identified the interaction of R2TP/PFDL with the TSC1–TSC2 complex in both TAP-MS and FLAG CoIP, we sought to strengthen our confidence in this important interactor by reciprocal purification. For this purpose, and to identify additional interactors of the TSC1–TSC2 complex, TAP-MS was performed on all three subunits (Fig. 3). Even though well-known interactor Rheb could not be identified from purification of any subunit of the TSC1–TSC2 complex, TAP-MS was nonetheless successful as evidenced by the detection of G-proteins RRAGA/RagA, RRAGC/RagC and Ragulator complex component LAMTOR1 which have been shown to anchor TSC1–TSC2 to the lysosome in the absence of amino acids (Fig. 2d)[41]. Purifications of the TSC1–TSC2 complex also corroborated the association to R2TP/PFDL as a number of subunits were detected. In contrast to the aforementioned interaction with snoRNPs which relies on NUFIP1 and ZNHIT6, no obvious protein candidate has been identified in TSC1–TSC2 purifications that could act as a bridging factor between the two complexes. Surprisingly, interactomes of TSC1 and TBC1D7 seem to have a lot more in common with each other than with TSC2. Indeed, although all TSC1–TSC2 complex subunits have managed to copurify RUVBL1 and RUVBL2, which are not exclusive to R2TP/PFDL, other R2TP/PFDL component have solely been detected in TSC1 and TBC1D7 purifications. Likewise, TSC2 purifications (which yielded 71.4% sequence coverage of the bait protein itself) recovered surprisingly few endogenous TSC1 (29.2% coverage) and TBC1D7 (6.7% coverage). This may be indicative of the existence of a TSC1–TBC1D7 subcomplex that

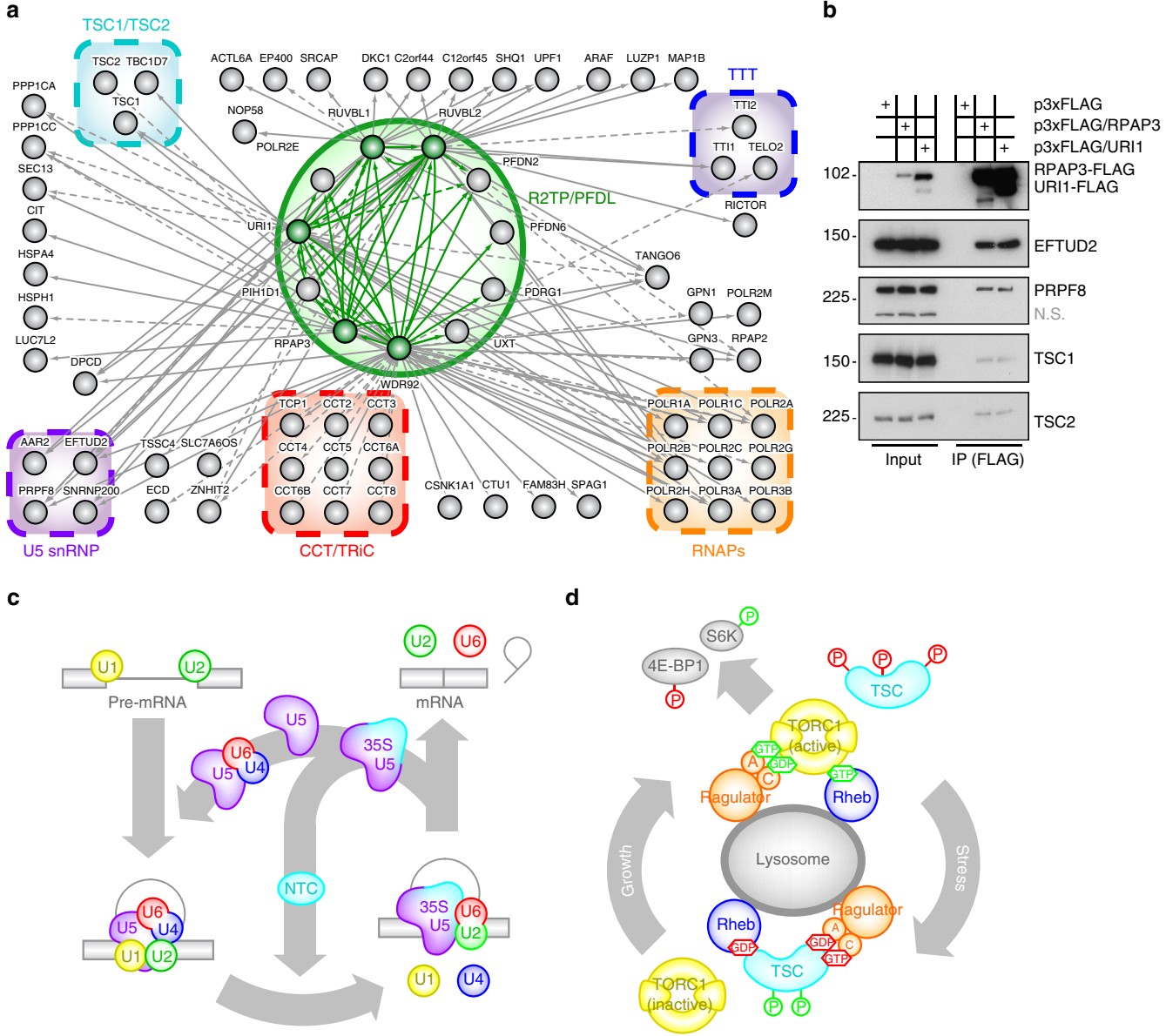

**Figure 2 | Affinity purification of R2TP/PFDL subunits reveals interactors. (a)** Diagram of the network of high-confidence interactions formed around the R2TP/PFD complex. Solid lines denote interactions with a FDR lower than 0.1 while dashed lines are interactions of relevance in this network with FDR scores higher than 0.1, but lower than 0.2 (Supplementary Data 1). Green-coloured nodes are tagged subunits used in this experiment. **(b)** CoImmunoprecipitation (CoIP) of FLAG-tagged subunits of the R2TP/PFD-like complex (RPAP3 and URI1) in HeLa S3 cells. Various Western blots were made to detect endogenous or recombinant proteins, as marked to the right. **(c)** Outline of the splicing cycle with emphasis on the recycling steps of the U5 snRNP and its reintegration within the U4/U6.U5 tri-snRNP (inner loop). **(d)** Regulation of the TORC1 kinase complex by TSC1–TSC2 (annotated here as 'TSC'). Guanine nucleotides (GTP, GDP) and phosphate groups (P) are indicated in green or red, depending on whether they have an activating or inhibitory effect on their associated proteins, respectively.

could be the result of a stepwise assembly mechanism or a novel mode of regulation for the complex. In accordance with this hypothesis, free pools of TSC2 have been observed *in vivo* and it has furthermore been shown that TBC1D7 can interact with TSC1 independently of TSC2 (ref. 40).

**ZNHIT2 is a cofactor of R2TP/PFDL that targets the U5 snRNP.** Although seemingly unrelated to TSC1–TSC2, the U5 snRNP is yet another interactor of R2TP/PFDL that was identified in our TAP-MS experiments (Fig. 2a, Supplementary Data 1). We next decided to map the protein interaction network of this ribonucleoprotein complex with an emphasis on factors possibly

involved in its biogenesis and/or recycling. To do so, we chose to perform TAP-MS on subunits of the U5 snRNP that have been reported to be transient component of non-catalytic complexes, namely AAR2 (refs 42–44) and CD2BP2 (ref. 45).

Purification of these subunits of U5 snRNP revealed the presence of factors associated with the R2TP/PFDL complexes, ZNHIT2 and SLC7A6OS. In the case of SLC7A6OS, the distantly related ortholog of yeast Iwr1, it came as a surprise that the protein was coupled more tightly with the U5 snRNP than nuclear RNA polymerases, for which Iwr1 had been demonstrated to act as an import factor[33]. ZNHIT2 has already been observed in conjunction with R2TP/PFDL[7,18,19] and it seemed reasonable that this interaction might confer specificity

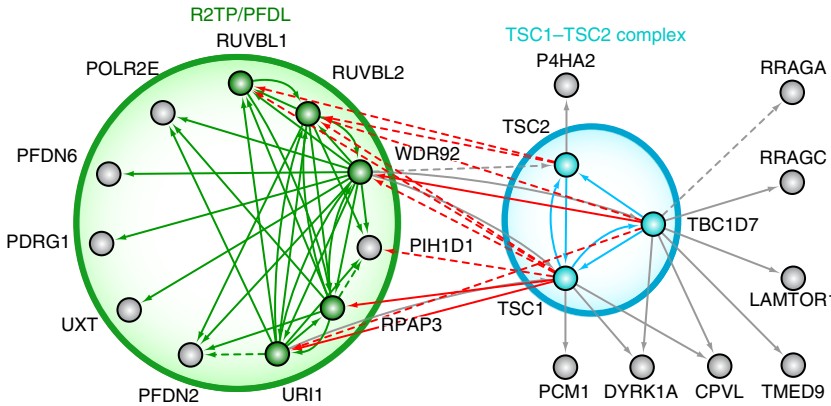

**Figure 3 | R2TP/PFDL is a major interactor of the TSC1–TBC1D7 subcomplex.** Diagram of the network of high-confidence interactions formed around the TSC1–TSC2 complex. Solid lines denote interactions with a FDR lower than 0.1 while dashed lines are interactions of relevance in this network with FDR scores higher than 0.1, but lower than 0.2 (Supplementary Data 1). Red arrows are R2TP/PFD-like subunits that copurified in TSC1–TSC2 complex purifications. Coloured nodes are tagged subunits used in this experiment.

towards a particular RNP complex, in a manner similar as ZNHIT6/BCD1 and ZNHIT3 for box C/D snoRNPs[9,14,15], but its exact function has nonetheless remained elusive to this day. Reciprocal TAP-MS, as well as proximity-dependent biotin identification (BioID-MS; Supplementary Data 2) of ZNHIT2 revealed substantial amounts of both the R2TP/PFDL complex and the U5 snRNP (Fig. 4a, Supplementary Data 1), suggesting that it may act as a bridging factor between the two.

This hypothesis was tested by CoIP experiments in cell treated with siRNAs directed against ZNHIT2 (Fig. 4b). FLAG-RPAP3 and FLAG-URI1 were independently purified and in both cases, association with U5 snRNP components EFTUD2 and PRPF8 was confirmed. However, when combined with ZNHIT2 knockdown, R2TP/PFDL subunits copurified substantially less U5 snRNP proteins as compared to cell treated with non-specific siRNAs, supporting a function of ZNHIT2 as a bridging factor.

The burgeoning network that began to take shape allowed the identification of a few new interactors of the U5 snRNP (Fig. 4a, Supplementary Data 1). EAPP, ECD, NCDN and TSSC4 are such proteins that copurified with U5 snRNP subunits, as well as with SLC7A6OS and ZNHIT2. As expected, reciprocal purification of EAPP, ECD and TSSC4 did confirm their interaction with the U5 snRNP. Hutchins et al. also noted the presence of EAPP, ECD, NCDN, TSSC4 and ZNHIT2 with purified murine PRPF8/Prp8 (ref. 46). Although these interactions were not investigated further, these results can be repurposed as an independent validation of our own data.

It should also be noted that purification of most of the aforementioned factors have yielded subunits of the NineTeen Complex or associated proteins (Fig. 4a, Supplementary Data 1). The observed form of U5 snRNP could thus be the NTC-associated, so-called '35S U5 snRNP' that materializes following spliceosome activation[36]. How the 35S U5 snRNP is retroconverted into the NTC-less, 20S form that is integrated within the U4/U6.U5 tri-snRNP is still a matter of debate. Association of these proteins with the 35S U5 snRNP may therefore point to a potential role in recycling rather than de novo biogenesis.

**The zinc finger HIT domain mediates interaction with RUVBL2.** ZNHIT2 is a member of a conserved family of zinc finger HIT (zf-HIT) domain-containing proteins (Fig. 5a,b). The zf-HIT domain consists of two anti-parallel β sheets followed by two α helices packed against the second β sheet. Histidine

and cysteine residues within a conserved CCCC–CCHC motif coordinate two zinc ions in an interleaved, cross-brace manner. The solution structure of the zf-HIT domain of ZNHIT2 has been resolved[47] and surface analysis revealed that although there is a hydrophobic patch commonly present at the surface of the zf-HIT domain, no large conserved patch of positively charged residues, characteristic of nucleotide binding zinc fingers, can be found. Furthermore, the tertiary structure of the domain resembles that of treble clef domains B-box, RING and PHD, all of which mediate protein–protein interactions.

A feature of known ZNHIT proteins is that they are often observed in complexes that also contain RUVBL1 and RUVBL2. As mentioned previously, ZNHIT6 and ZNHIT3 are snoRNP biogenesis factors that interact with NUFIP1 and the AAA+ ATPases[9,14,15]. ZNHIT1 and INO80B/ZNHIT4 are components of chromatin remodelling complexes SRCAP and INO80, respectively, both of which also harbour a RUVBL1 and RUVBL2 module[7].

Given these indicative structural and interaction data, we tested whether the zf-HIT domain of ZNHIT2 could also sustain RUVBL1 and/or RUVBL2 binding and, more generally, if binding to these AAA+ ATPases was a conserved trait for all zf-HIT domain-containing proteins. An in vitro pull-down assay was performed with GST-tagged ZNHIT2 and polyhistidine-tagged RUVBL1 and RUVBL2 (Fig. 5c). In presence of ATP, ZNHIT2 showed affinity towards RUVBL2, but not RUVBL1. When ATP was replaced by ADP, binding was increased, but substitution with a non-hydrolyzable analogue, γ-S-ATP, conversely resulted in decreased interaction. These results essentially mimic those obtained for ZNHIT6/BCD1 by McKeegan et al.[9] A mutant of ZNHIT2 where the zf-HIT domain was absent (deleted region indicated in Fig. 5a) was then used to show that the binding of RUVBL2 is indeed mediated by this protein domain. The experiment was then extended to other human ZNHITs (Fig. 5d) and we observed that each of these proteins interact preferentially with RUVBL2 but, when their respective zf-HIT domain are deleted, binding is either lost or greatly reduced.

**Splicing is affected by ZNHIT2 and RUVBL2 expression levels.** Based on its interaction with the U5 snRNP, a logical assumption would be that the protein may have a role in pre-mRNA splicing. To assess whether ZNHIT2, or its binding partner RUVBL2, impact splicing, total RNA was purified from HEK 293 cells

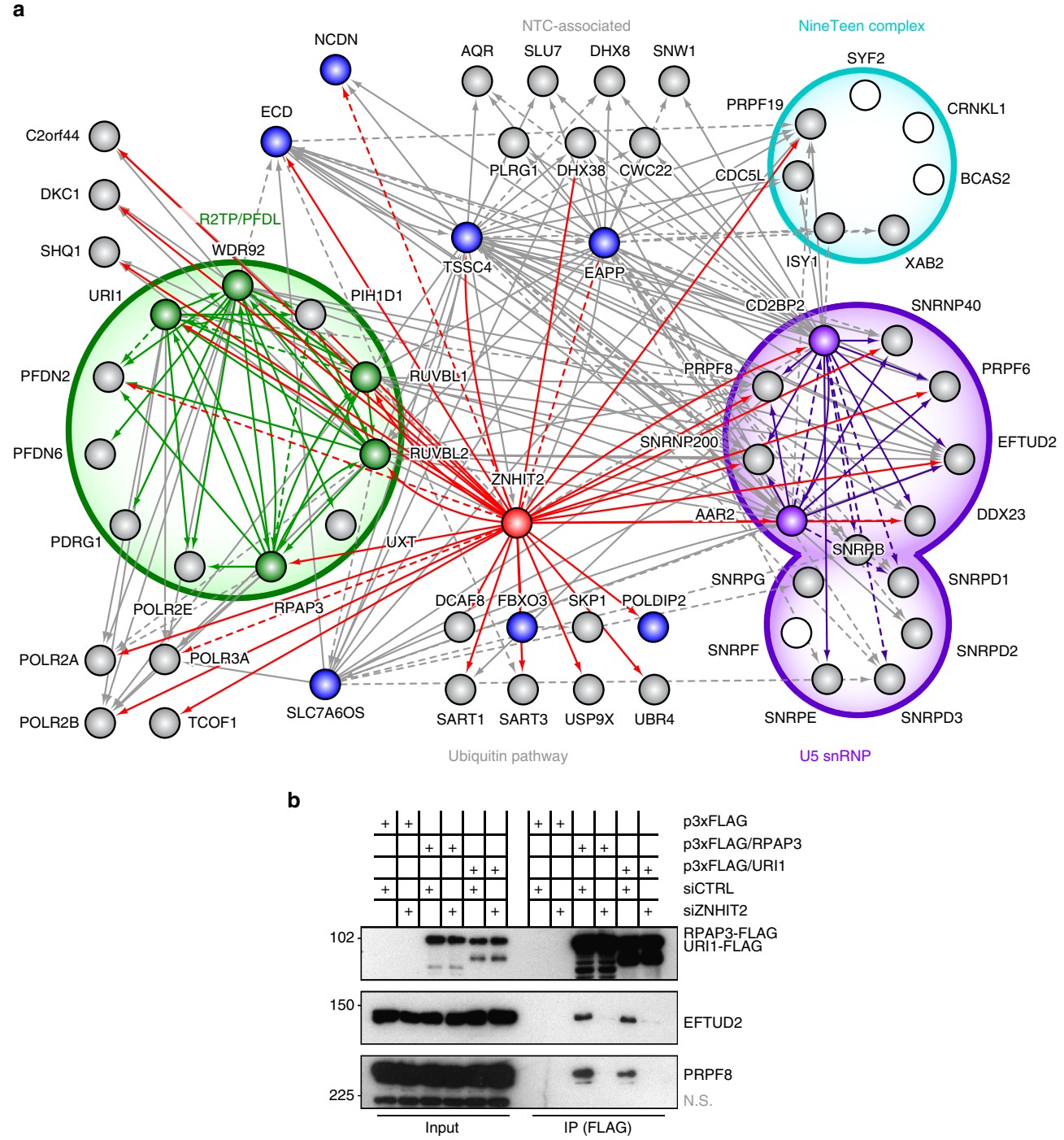

**Figure 4 | ZNHIT2 is a cofactor of R2TP/PFDL that targets the U5 snRNP. (a)** Diagram of the network of high-confidence interactions formed around ZNHIT2. Solid lines denote interactions with a FDR lower than 0.1 while dashed lines are interactions of relevance in this network with FDR scores higher than 0.1, but lower than 0.2 (Supplementary Data 1). Red arrows are target proteins identified in the ZNHIT2 purification. Coloured nodes are tagged subunits used in this experiment. White nodes are known protein complex subunits that were not detected. **(b)** CoImmunoprecipitation (CoIP) of FLAG-tagged subunits of the R2TP/PFD complex (RPAP3 and URI1) in HEK-293 cells following transfection with either non-specific (siCTRL) or ZNHIT2-targetting siRNAs (siZNHIT2). Various Western blots were made to detect endogenous or recombinant proteins, as indicated on the right side.

transfected with either non-specific siRNAs or siRNAs directed against ZNHIT2 and RUVBL2. We then assayed a set of randomly selected alternatively spliced events (ASE) from previous investigations[48–51]. We chose to monitor alternative splicing events as opposed to constitutive ones because ASEs are generally more sensitive to alterations in splicing efficiency. RT-PCR was performed on our high-throughput platform with

primers designed to amplify a relatively small region of no more than 500 bp spanning the ASEs (Fig. 6a). Relative abundance of amplicons was measured by capillary microfluidic fractionation and converted to a ratio of the molarity of the long product divided by the combined molarities of the long and short products ($\Psi$ value). Out of the 39 monitored ASEs, six showed significant modulation in their splicing profiles following

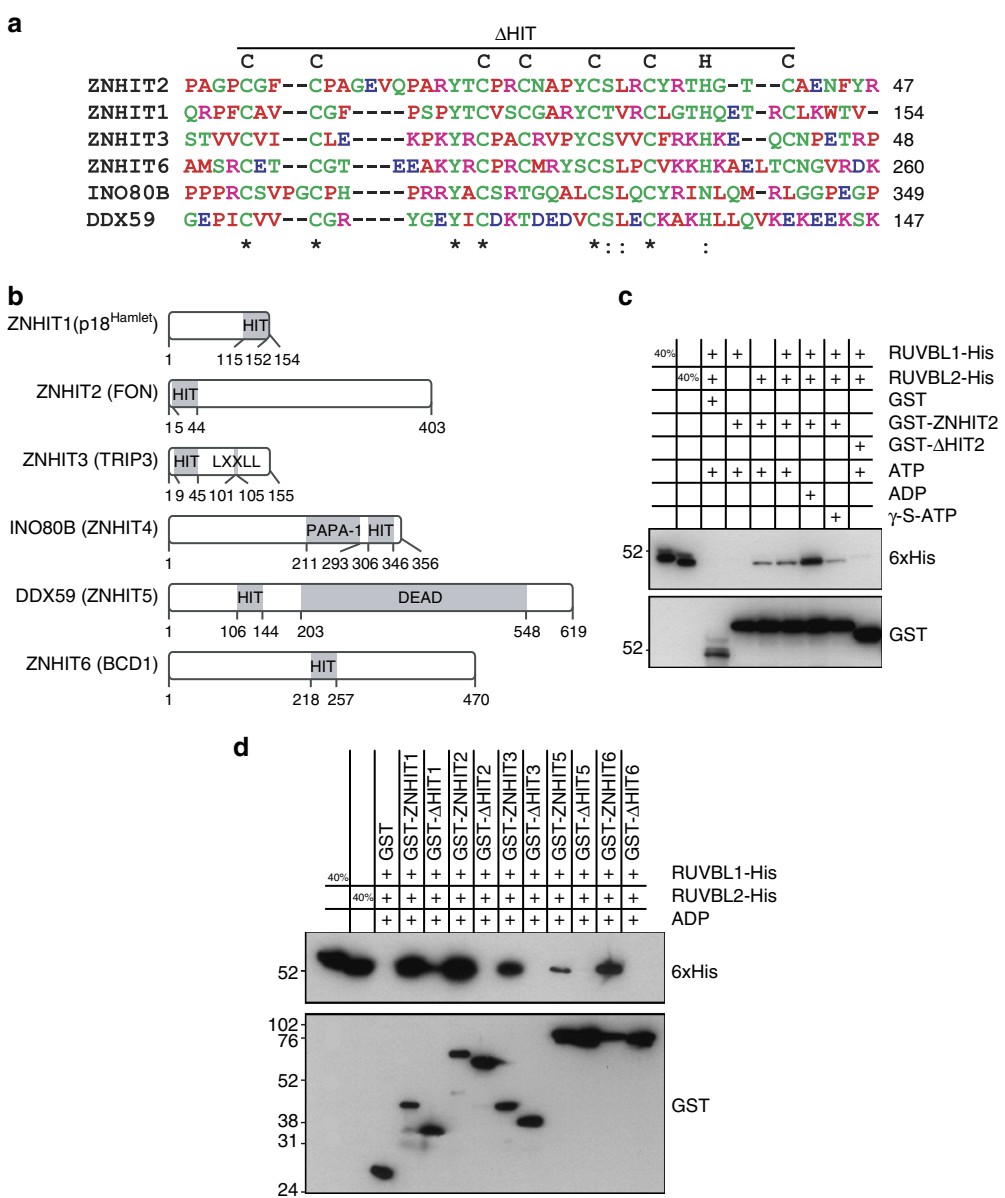

**Figure 5 | The zinc finger HIT domain mediates interaction with RUVBL2.** (**a**) Multiple sequence alignment of the zf-HIT domain in all six members of the human ZNHIT family in accordance with He *et al.*[47] Red residues are small, hydrophobic, aromatic; blue are acidic; magenta are basic; and all other residues are green. Residue similarity and identity are represented by colon (:) and asterisk (*), respectively. Zinc coordinating cysteine and histidine residues (or corresponding positions) and region deleted in ΔHIT mutants are indicated overhead. (**b**) Linear representation of protein domain architecture within members of the human ZNHIT family. Residues delineating each domain are indicated below. HIT, zinc finger HIT domain; LXXLL, coactivator LXXLL nuclear receptor recognition motif; PAPA-1, PAPA-1 homology region; DEAD, DEAD-box RNA helicase domain. (**c**) *In vitro* GST pull-down assays of GST-ZNHIT2 (either full-length or lacking the zf-HIT domain) with RUVBL1-His and/or RUVBL2-His in presence of ATP, ADP or non-hydrolyzable ATP analogue, γ-S-ATP. (**d**) *In vitro* GST pull-down assays of various GST-tagged ZNHIT proteins (either full-length or lacking the zf-HIT domain) with RUVBL1-His and RUVBL2-His in presence of ADP.

ZNHIT2 knockdown, four of which saw similar variations when RUVBL2 was targeted (Fig. 6b).

**ZNHIT2 and RUVBL2 regulate composition of the U5 snRNP**. As mentioned previously, the R2TP/PFDL complex is involved in the biogenesis of a number of RNP complexes with the assistance of ZNHIT3 and ZNHIT6. We next sought to determine if a similar scenario could be at play with ZNHIT2 in the assembly of the U5 snRNP. To do so, we monitored changes in U5 snRNP protein composition following ZNHIT2 siRNA-mediated knockdown using quantitative mass

spectrometry with stable isotope labelling (SILAC) (Fig. 7a). Untransfected HEK 293 cells were grown in medium supplemented with unlabelled (light) amino acids while cells transfected with expression vectors for FLAG-tagged subunits of the U5 snRNP (either PRPF8 or EFTUD2) and a non-specific control siRNA were cultured in a medium containing deuterium-labelled lysine and [13]C-labelled arginine (medium). The unlabelled purified proteins are those that interact non-specifically with anti-FLAG antibody-coupled beads and therefore serve as a purification control. Proteins purified from medium-labelled conditions are enriched in U5 snRNP assembled in control

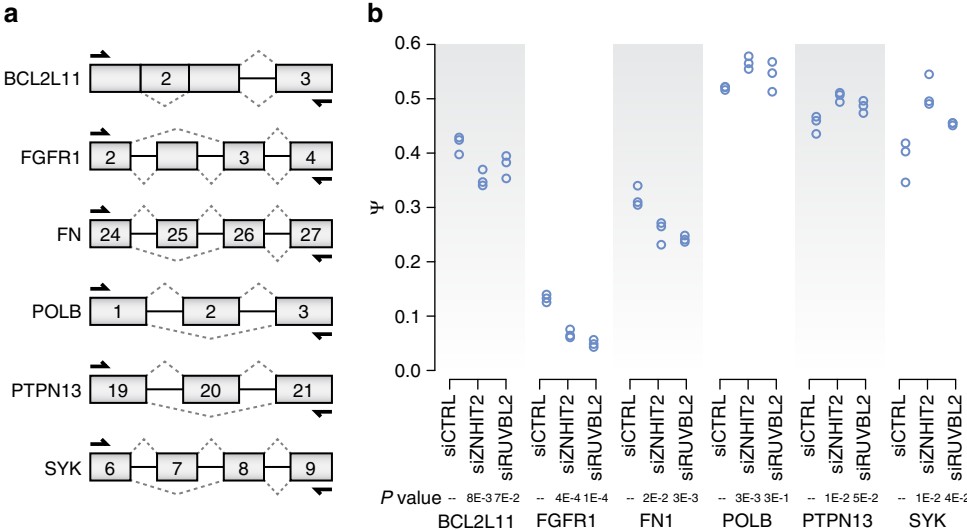

**Figure 6 | Pre-mRNA splicing is affected by ZNHIT2 and RUVBL2 expression levels.** (**a**) Diagram of alternative splicing events monitored in this experiment. Primer pairs targeting alternative splicing events are shown. (**b**) Significant splicing shifts ($P < 0.05$) following ZNHIT2, or RUVBL2 knockdown. PSI ($\Psi$) values for three biological replicates of six alternative splicing events following transfection of a non-targeting siRNA, ZNHIT2 knockdown and RUVBL2 knockdown are indicated.

conditions. Experimental conditions are enacted in $^{13}$C and $^{15}$N doubled-labelled arginine and lysine (heavy)-supplemented media where cells are double transfected with the same FLAG expression vectors and a siRNA directed against ZNHIT2. MS data analysis by MaxQuant[52] rendered light/medium/heavy SILAC ratios of proteins purified from all three conditions and SILAC ratios were plotted on a two-dimensional logarithmic graph where the enrichment of U5 snRNP-associated proteins is found at the right extremity of the x axis (medium/light ratio) and proteins lost or enriched following ZNHIT2 siRNA treatment are found at the top and bottom of the y axis, respectively (medium/heavy ratio).

In both PRPF8- and EFTUD2-based purifications, we found a significant decrease in association of RUVBL1 and RUVBL2 (Fig. 7b,c, Supplementary Data 3 and 4), consistent with the hypothesis that ZNHIT2 mediates association of the U5 snRNP with the R2TP/PFDL chaperone complex. Surprisingly, no other R2TP/PFDL subunits were detected in these purifications which could presumably be explained by the multimeric nature of RUVBL1 and RUVBL2. Indeed, each protein is believed to be present in six copies in R2TP/PFDL which makes them more readily detectable by MS. Alternatively, ZNHIT2 could also interact with the RUVBL1/RUVBL2 complex regardless of whether or not it is integrated within the framework of R2TP/PFDL. In this case, the putative chaperoning function would therefore be attributed directly to the AAA + proteins as was found to be the case for snoRNPs[12]. Although knockdown of ZNHIT2 diminished the association of RUVBL1 and RUVBL2 with the U5 snRNP, no significant disparity was found in any of the core components of the U5 snRNP, which could be due to residual association of RUVBL1 and RUVBL2 that could still achieve chaperoning function. Indeed, the decrease in ZNHIT2 that copurified with the U5 snRNP, which is the direct result of its reduced expression following RNAi, was clearly more pronounced than that of RUVBL1 and RUVBL2. It is possible that the ZNHIT2 knockdown, although considerable, may not have been enough to result in an observable effect on U5 snRNP composition. There may also be an alternative pathway for the R2TP/PFDL complex to interact with the U5 snRNP. The best candidate as a surrogate bridging factor is ECD which

has already been shown to interact directly with R2TP/PFDL, more specifically its subunit PIH1D1/Nop17 in a phosphorylation-dependent manner[53,54] and was observed in our own purifications of the U5 snRNP (Fig. 4a, Supplementary Data 1). In support of this hypothesis, we observed that knockdown of ECD, either by itself or in conjunction with ZNHIT2, can likewise affect U5 snRNP protein composition (Supplementary Fig. 1, Supplementary Data 5 and 6).

We next opted to repeat the experiment with a siRNA targeting RUVBL2 mRNA directly to determine whether the AAA + ATPase is indeed implicated in U5 snRNP complex formation. Association of RUVBL2 with either EFTUD2 or PRPF8 was greatly compromised attesting to the efficiency of the knockdown (Fig. 7e,f, Supplementary Data 7 and 8). Significant changes were also observed in U5 snRNP protein composition. For example, EFTUD2-FLAG purifications yielded less PRPF8/Prp8/U5-220K, SNRNP200/Brr2/U5-200K and DDX23/Prp28/U5-100K following RUVBL2 knockdown. Similarly, we observed considerably reduced amounts of PRPF6/Prp6/U5-102K, SNRNP40/SPF38/U5-40K and CD2BP2/Snu40/U5-52K that copurified with PRPF8-FLAG. The fact that different subunits seemed to be perturbed depending on whether EFTUD2 or PRPF8 is being used for U5 snRNP purification may point to an assembly pathway involving distinct intermediate complexes as was observed for the R2TP/PFDL-dependent biogenesis of RNAPII (ref. 20), with EFTUD2 interacting intimately with one subcomplex and PRPF8 associating more closely with the other.

Since the RUVBL1/RUVBL2 module is an integral part of a number of chromatin remodelling factors as well as having a role in the biogenesis of nuclear RNAPII and the ribosome (notably through snoRNP assembly), there was a distinct possibility that the discrepancy observed in U5 snRNP composition following RUVBL2 knockdown was not the result of interference with the assembly process, but rather a diminished expression of the various subunits. To address this possibility, expression of a subset of U5 snRNP subunits were assessed following RNAi by western blot (Fig. 7b). No variation was observed in the expression of PRPF8, EFTUD2, SNRNP200, CD2BP2, PRPF6, DDX23, SNRNP40 or AAR2 when cells were treated with either non-specific control siRNA, ZNHIT2- or

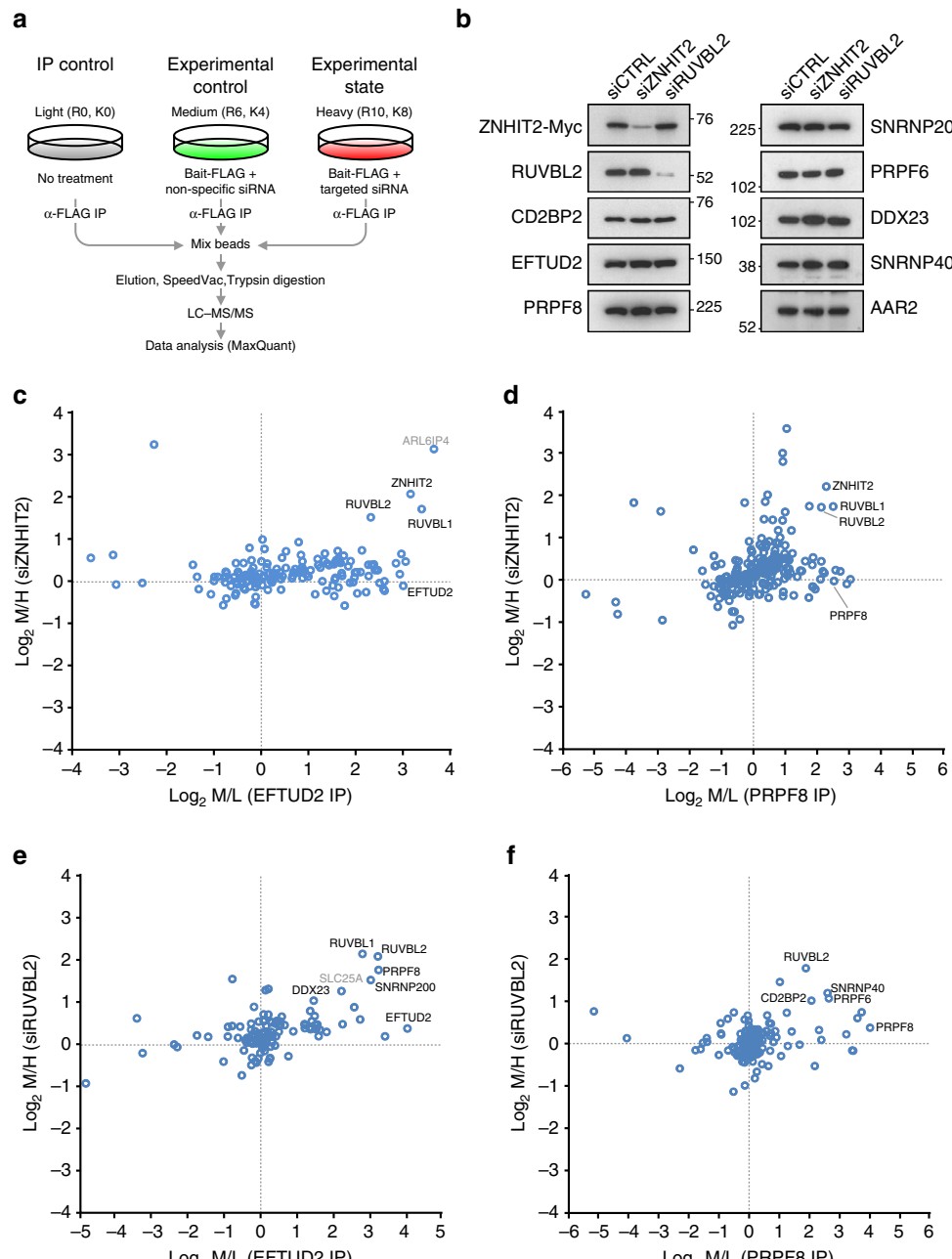

**Figure 7 | ZNHIT2 and RUVBL2 regulate the composition of the U5 snRNP. (a)** Workflow of triple SILAC experiments. **(b)** Expression assessment of various U5 snRNP subunits following treatment with siZNHIT2 or siRUVBL2. **(c–f)** Interpretation of the SILAC data on scatterplots. Intensity ratio of medium-labelled proteins over unlabelled proteins (M/L) on the X axis denotes IP enrichment. Intensity ratio of medium-labelled proteins over heavy-labelled proteins (M/H) on the Y axis represents experimental variations. Marked proteins in the upper right quadrant ($X>1$, $Y>1$) are proteins enriched in the IP that were less abundant following siRNA treatment. Position of the protein used for IP is shown in all experiments.

RUVBL2-targeting siRNAs. Efficiency of the knockdown was assessed directly in the case of RUVBL2 and indirectly through transient expression of Myc-tagged ZNHIT2 (as we have yet to find a reliable ZNHIT2 antibody). Together, these results indicate that ZNHIT2 and RUVBL1/RUVBL2 interact to regulate the integrity of the U5 snRNP complex.

## Discussion

We report here the identification of two interactors of the R2TP/PFDL cochaperone complex, namely the U5 snRNP

and the TSC protein complex. In the case of U5 snRNP, the protein interaction network that has been drawn around this spliceosome component led to the identification of numerous interaction partners including ZNHIT2 which was shown, through extensive experimental evidence, to act as a bridging factor between the R2TP/PFDL cochaperone complex and the U5 snRNP. A broader role for the zf-HIT domain, a key feature shared by six different proteins encoded by the human genome, was uncovered as a nucleotide-dependent RUVBL2 binding module. We have gone further to demonstrate that RUVBL1/RUVBL2 can alter U5 snRNP composition although it is not yet

clear whether this effect is solely dependent on the activity of the AAA+ ATPases or can also be imputed to other components of R2TP/PFDL. Despite clear involvement of RUVBL1/RUVBL2 in the assembly of the U5 snRNP complex, we are cautious not to assume that this implies a role for the AAA+ ATPases in *de novo* biogenesis of the ribonucleoparticle as it could just as likely play a part in U5 snRNP recycling, a process that is still poorly understood.

Most studies published so far on R2TP/PFDL have assumed that the complex is involved in *de novo* multi-protein or protein–RNA complex assembly. While this hypothesis seems plausible given the classical role of chaperones in assisting nascent proteins and protein complexes attain their native state, it does however suffer from a few caveats. Firstly, the composition of R2TP/PFDL varies tremendously between organisms with the *S. cerevisiae* counterpart missing WDR92/Monad and the prefoldin-like module entirely. Moreover, genes encoding subunits specific to R2TP (Tah1 and Pih1) are non-essential in yeast as demonstrated by the use of deletion mutants. This casts doubt on whether R2TP/PFDL proteins are required for the biogenesis of such critical complexes like nuclear RNA polymerases, PIKKs, L7Ae RNPs or the U5 snRNP. A likely explanation could be that, while not necessary in optimal conditions, the cochaperone complex might help the cell optimize assembly processes under stress. In this regard, it has been noted that snoRNP complex composition is mostly unaffected by yeast R2TP deletion mutants but that disruption in assembly arises when cells are grown under stress condition[10]. More recently, R2TP localization was shown to be affected by mTOR inhibitor treatment and that snoRNP biogenesis was consequently regulated by nutrient availability[55]. This is not the first time that a role has been proposed for R2TP/Prefoldin-like as an effector of the mTOR pathway. In fact, one of the earliest reports on this complex posited that R2TP/PFDL controls expression of a subset of genes in a nutrient-sensitive manner[56]. While no direct link has since been found between R2TP/PFDL and targeted gene expression, the idea of R2TP as an effector of mTOR could hypothetically be a way of explaining lesser known aspects of nutrient-availability response. Indeed, despite translation inhibition being the most prominent feature of mTOR-mediated regulation, other aspects of cell biology are likewise affected, including rRNA transcription and maturation[57], ATM activity[58] and telomerase activity[59], all of which could theoretically be regulated by targeting the chaperoning activity of R2TP/PFDL.

Pre-mRNA splicing has also been shown to be hindered in stress conditions, including amino acid starvation in yeast[60,61]. With introns being so tightly conserved in ribosomal protein-coding genes, spliceosome inhibition is yet another way of ultimately blocking translation based on nutrient availability. While the exact mechanism through which this regulation takes place is not known, a similar repression of pre-mRNA splicing has also been observed following heat-shock[62–65]. Under such conditions, the U5 snRNP appears unable to integrate within the U4/U6.U5 tri-snRNP or the fully assembled spliceosome. It is striking that U5 snRNP obtained from R2TP/PFDL and ZNHIT2 affinity purifications was likewise unassembled and showed only trace of proteins commonly associated with other snRNPs. It may be that this particular state of U5 snRNP is targeted by a number of different factors (which may include ZNHIT2 and R2TP/PFDL and any of the U5 snRNP-associated factors identified in this manuscript) under different stress conditions as a way of regulating splicing at large.

The extensive connectivity of R2TP/PFDL with the mTOR pathway was only strengthened by our own identification of the seemingly direct interaction of the cochaperone with the TSC1–TSC2 complex. This physical association could help explain some of the phenotypic linkage between the two complexes. Indeed, TSC1–TSC2 is notorious for its involvement in tumour growth and a number of studies have uncovered such a link for R2TP/PFDL. Most have focused on RUVBL1 and RUVBL2 (ref. 66) and as such cannot be attributed solely to the R2TP/PFDL complex as the carcinogenic activity could just as easily be ascribed to the chromatin remodelling complexes containing these AAA+ ATPases or oncogenic transcription factors c-Myc[67] and β-catenin[68], both of which require RUVBL1 and RUVBL2 as cofactors. However, some subunits specific to R2TP/PFDL, namely URI1 and UXT, were shown to be amplified, overexpressed or otherwise linked to tumour progression in prostate, ovarian, cervical and endometrial cancer, hepatocellular carcinoma and multiple myeloma[69–76]. The exact function of R2TP/PFDL in the case of TSC1–TSC2 remains unknown but protein complex composition was assessed following siRNA-mediated knockdown of R2TP/PFDL subunit WDR92 and no obvious changes were observed (Supplementary Fig. 2, Supplementary Data 9). We suspect that interaction with R2TP/PFDL could provide an alternative pathway for TSC1–TSC2 to regulate mTOR activity through chaperoning of the latter. More broadly, this association could be further evidence of a function for the cochaperone complex in regulating anabolic processes by targeting the assembly of various key proteins complexes.

The existence of R2TP/PFDL has only been revealed a few years ago. Conversely, the cochaperone has already been linked to critical protein complexes involved in fundamental cellular processes that govern gene expression at all levels from transcription to translation. While we have yet to fully appreciate the complete repertoire of protein complexes targeted by its chaperoning activity and understand the specifics of its mode of action, a role for R2TP/PFDL as a major regulator of anabolic processes is beginning to emerge and will undoubtedly spark interest in this exciting area of research.

## Methods

**DNA constructs.** All tagged constructs used in TAP-MS, BioID-MS, SILAC, CoIP and GST pulldown experiments were produced by PCR amplification of cDNA using Mammalian Gene Collection clones (GE Healthcare) or pRK7-FLAG-TSC1 and pRK7-FLAG-TSC2 plasmids (kindly provided by Dr John Blenis) as template. The resulting fragments were subsequently cloned into pMZI, pDEST-pcDNA5-BirA*-FLAG C-term (kindly provided by Dr Anne-Claude Gingras), p3xFLAG-CMV-14 (Sigma), pcDNA3-Myc, pGEX-4T-1 (GE Health-care) or pET-23a+ (Novagen) expression vectors (see Supplementary Table 1 for a list of all plasmids created for this article and the strategy used to create them). Deletion mutants for ZNHIT proteins were produced by site-directed mutagenesis to precisely remove regions corresponding to the zf-HIT domain as annotated in Fig. 4.

**Cell culture.** HeLa S3 and HEK 293 cells were obtained from ATCC, tested for mycoplasma and cultured in DMEM media supplemented by 10% fetal bovine serum and 2 mM glutamine. Flp-In T-REx 293 cell line was obtained from Thermo Scientific. For siRNA transfections, cells were seeded and grown overnight to a confluence of about 20%, and then transfected using Lipofectamine 2000 (Thermo Scientific) according to the manufacturer's instructions. SMARTpool siGenome siRNAs (GE-Healthcare) targeting human ZNHIT2 (M-008955-00), RUVBL2 (M-012299-00), ECD (M-019678-01) and WDR92 (M008669-01) were employed at a concentration of 100 nM. Non-targeting siRNA (D-001210-01) was included as a control. After 72 h of siRNA transfection, cells were collected for CoIP. For DNA transfections, cells were seeded and grown overnight to a confluence of about 80%, and then transfected using Lipofectamine 2000 (Thermo Scientific) according to the manufacturer's instructions and harvested 48 h later for CoIP. For TAP-MS experiments, stable cell lines were obtained as detailed previously[18] by growing transfected EcR 293 cells (a HEK 293 cell line expressing the EcR ecdysone receptor) in 30 µg ml$^{-1}$ bleocin, 300 µg ml$^{-1}$ G418 (Thermo Scientific) selective medium. For BioID-MS experiments, stable cell line populations were produced by transfecting pDEST-pcDNA5-BirA-FLAG vectors and pOG44 (Thermo Scientific) Flp-recombinase expression vector in a 1:10 ratio into Flp-In T-Rex 293 cells (Thermo Scientific) and selecting for resistance in 200 µg ml$^{-1}$ hygromycin. For SILAC experiments, HEK 293 cells were grown for three weeks in minimal DMEM supplemented with 10% dialyzed

fetal bovine serum (Wisent), 0.8 mM lysine and 0.4 mM arginine. Standard isotopes $^{12}C_6,^{14}N_2$ lysine and $^{12}C_6,^{14}N_4$ arginine (Sigma) were used in light SILAC medium, whereas medium SILAC medium contained 4,4,5,5-D$_4$ lysine and $^{13}C_6$ arginine (Cambridge) and heavy SILAC medium was supplemented with $^{13}C_6,^{15}N_2$ lysine and $^{13}C_6,^{15}N_4$ arginine (Cambridge). An aliquot of 0.5 mM proline was added to all SILAC media to curb arginine to proline conversion.

**Tandem affinity purification coupled to mass spectrometry.** TAP-tagged proteins expression was induced with 3 μM ponasterone A (Thermo Scientific) for 48 h. One gram of collected cells were lysed in 2.5 ml of lysis buffer A (10 mM Tris–HCl pH 8, 0.34 M sucrose, 3 mM CaCl$_2$, 2 mM magnesium acetate, 0.1 mM EDTA, 1 mM DTT, 0.5% NP-40, 0.5 mM AEBSF and complete EDTA-free protease inhibitor cocktail (Roche)) using a tissue grinder (Wheaton). Following centrifugation (3,500 g, 15 min, 4 °C), the pellet was homogenized once more in 2.5 ml of lysis buffer B (20 mM HEPES pH 7.9, 1.5 mM MgCl$_2$, 150 mM potassium acetate, 3 mM EDTA, 1 mM DTT, 0.1% NP-40, 10% glycerol, 0.5 mM AEBSF and protease inhibitor cocktail). After centrifugation (15,000 g, 30 min, 4 °C) the chromatin-containing pellet was set aside and both supernatants containing soluble proteins were combined and spun once more (165,000 g, 1 h 30, 4 °C) and dialyzed overnight in 2 l of dialysis buffer (10 mM HEPES pH 7.9, 0.1 mM EDTA, 0.1 mM DTT, 0.1 M potassium acetate and 10% glycerol). The chromatin pellet was chopped into smaller bits with a scalpel in 2 ml of nuclease incubation buffer (150 mM HEPES pH 7.9, 1.5 mM MgCl$_2$, 150 mM potassium acetate, 10% glycerol, 0.5 mM AEBSF and protease inhibitor cocktail) and then subsequently ground to homogeneity in a glass tissue grinder (Kontes). The partially solubilized chromatin fraction was digested overnight in 150 U ml$^{-1}$ benzonase, 50 U ml$^{-1}$ RNase A and 10 U ml$^{-1}$ DNaseI. The next day, both soluble and chromatin fractions were cleared of insoluble material by centrifugation (20,000 g, 30 min, 4 °C) and incubated individually with 25 μl of IgG sepharose beads (GE Healthcare) for 1 h at 4 °C on a tube rotator. The resin and bound complexes were washed twice in 250 μl immunoprecipitation buffer (10 mM Tris–HCl, pH 8, 100 mM NaCl, 0.1% Triton X-100, 10% glycerol) and transferred to a Micro Bio-Spin Chromatography Column (Bio-Rad). Two volumes of 500 μl TEV buffer (10 mM Tris–HCl pH 8, 100 mM NaCl, 0.1% Triton X-100, 0.5 mM EDTA, 10% glycerol, 1 mM DTT) were used to rinse the column which was then sealed with 150 μl of TEV buffer and 15 U of AcTEV protease (Thermo Scientific). After an overnight incubation (at 4 °C on a tube rotator), the eluate was collected into another Micro Bio-Spin Chromatography Column containing 25 μl of calmodulin sepharose 4B beads (GE Healthcare). Two volumes of 300 μl of calmodulin binding buffer (10 mM Tris–HCl pH 8, 100 mM NaCl, 1 mM imidazole, 1 mM magnesium acetate, 2 mM CaCl$_2$, 0.1% Triton X-100, 10% glycerol, 10 mM β-mercaptoethanol) were used to ensure maximal recovery of protein complexes from the IgG sepharose column. The calmodulin resin-containing column was sealed with 0.8 μl of 1 M CaCl$_2$, and incubated 2 h at 4 °C on a tube rotator. The resin was washed twice in 500 μl calmodulin binding buffer and the proteins were eluted by successive addition of 100 μl and 150 μl of calmodulin elution buffer (10 mM Tris–HCl pH 8, 100 mM NaCl, 1 mM imidazole, 1 mM magnesium acetate, 2 mM EGTA, 10% glycerol, 10 mM β-mercaptoethanol). Trichloroacetic acid protein precipitation was performed to remove detergents from the samples and protein extracts were then resolubilized in 10 μl of a 6 M urea buffer. Proteins were reduced by addition of 2.5 μl of the reduction buffer (45 mM DTT, 100 mM ammonium bicarbonate) and incubating for 30 min at 37 °C, and subsequently alkylated with 2.5 μl of the alkylation buffer (100 mM iodoacetamide, 100 mM ammonium bicarbonate) and incubating for 20 min at 24 °C in the dark. Prior to trypsin digestion, 20 μl of water was added to reduce the urea concentration to 2 M. 12.5 ng of Trypsin Gold (Promega) was added to each sample. Protein digestion was performed at 37 °C for 18 h and stopped with 5 μl of 5% formic acid. Protein digests were dried down in a speed-vac and stored at −20 °C until liquid chromatography (LC)–MS/MS analysis. Prior to LC–MS/MS, protein digests were re-solubilized under agitation for 15 min in 10 μl of 0.2% formic acid. Desalting and cleanup of the digests was performed by C18 ZipTip pipette tips (Sigma). Eluates were dried down in a speed-vac and then resolubilized under agitation for 15 min in 10 μl of 2% acetonitrile, 1% formic acid.

**Proximity-dependent biotinylation (BioID).** The protocol for biotin identification of ZNHIT2 interactors was modified from Couzens et al.[77] BirA*-tagged protein expression and biotinylation was induced with 1 μg ml$^{-1}$ of tetracycline and 75 μM D-biotin. The next day, 0.1 g of collected cells were lysed in 2 ml of RIPA buffer (50 mM Tris–HCl pH 7.4, 150 mM NaCl, 0.5% sodium deoxycholate, 0.1% SDS, 1% NP-40, 1 mM EDTA, 1 mM DTT, 1 mM PMSF, complete EDTA-free protease inhibitor cocktail (Roche) and 250 U of benzonase) with a Branson digital sonifier S-450 (three 10 s bursts at 30% amplitude). Following centrifugation (12,000 g, 30 min, 4 °C), the supernatant was transferred to 35 μl streptavidin sepharose high performance beads (GE Healthcare) and incubated for 3 h at 4 °C on a tube rotator. The resin was washed three times in 1 ml RIPA buffer and three times in 50 mM ammonium bicarbonate pH 8.5 and resuspended in 100 μl of the latter. Proteins were digested on-beads by incubation with 0.5 μg sequencing grade modified trypsin (Promega) at 37 °C on a ThermoMixer (Eppendorf) at 600 r.p.m. for 18 h. Solubilized tryptic peptides were collected the next day following centrifugation (500 g, 1 min). Two cycles of resuspension/

centrifugation with 100 μl of HPLC-grade water were performed on the beads to increase peptide yield. Pooled fractions were reduced with 9 mM dithiothreitol (30 min, 37 °C) and alkylated in 17 mM iodoacetamide (20 min, room temperature in the dark). The samples were acidified with trifluoroacetic acid for desalting and removal of residual detergents on an Oasis MCX 96-well μElution Plate (Waters Corporation) following the manufacturer's instructions. After elution in 10% ammonium hydroxide/90% methanol (v/v), samples were dried with a speed-vac and reconstituted in 2% acetonitrile, 1% formic acid.

**Immunoprecipitation of FLAG-tagged proteins.** For CoIP experiments, HEK 293 or HeLa S3 cells were lysed in CoIP buffer (50 mM Tris–HCl, pH 7.6, 250 mM NaCl, 5 mM EDTA, 0.5% NP-40, 10 mM NaF, 0.2 mM sodium orthovanadate, 1 mM DTT, protease inhibitor cocktail) for 30 min at 4 °C on a tube rotator. Lysates were cleared of insoluble material by centrifugation (16,000 g, 10 min, 4 °C) and protein concentration was determined using Protein Assay Dye Reagent (Bio-Rad). One milligram of proteins were added to 10 μl of Anti-FLAG M2 magnetic beads (Sigma; M8823) that had been washed twice beforehand in TNET buffer (50 mM Tris–HCl, pH 7.6, 100 mM NaCl, 5 mM EDTA, 0.5% Triton X-100). Following a 2 h incubation at 4 °C on a tube rotator, the beads were washed four times in 1 ml of CoIP buffer and the bound complexes were eluted in sample buffer and subjected to immunoblot analysis.

For SILAC experiments, HEK 293 cells transfected with both expression vectors for FLAG-tagged constructs and siRNAs were grown in media containing isotope-labelled amino acids, harvested and frozen in liquid nitrogen. The cell pellets were thawed and membranes were disrupted in lysis buffer (25 mM HEPES pH 7.9, 100 mM KCl, 2 mM EDTA, 0.1% NP-40, 10% glycerol, 0.1 mM DTT, protease inhibitor cocktail) for 10 min at 4 °C on a tube rotator. Following a second liquid nitrogen freeze-thaw cycle, the soluble phase was purified by centrifugation (16,000 g, 10 min, 4 °C) and protein concentration was determined. Lysates from heavy-, medium-labelled and unlabelled cultures were incubated separately with Anti-FLAG M2 magnetic beads according to a proportion of 10 μl of beads (washed twice in lysis buffer) per 1 mg of proteins. The resin was incubated for 2 h at 4 °C on a tube rotator, washed four times in 1 ml of lysis buffer and then twice in 1 ml of rinsing buffer (50 mM ammonium bicarbonate, 75 mM KCl, pH 8.0). Beads from all three labelling conditions were then pooled and protein complexes were eluted by three consecutive incubations with 150 μl of fresh elution buffer (10% NH$_4$OH, pH 12.0) for 15 min at 4 °C on a tube rotator and the eluate was dried in a speed-vac. Trypsin digestion was performed as in TAP-MS experiments.

***In vitro* GST pulldown.** Recombinant proteins were produced by transforming One Shot BL21 Star DE3 cells (Thermo Scientific) with either pGEX or pET vectors described above. Bacteria were induced for 3 h with 0.5 mM IPTG (Sigma) and subsequently disrupted in lysis buffer (for 6 × His; 50 mM NaH$_2$PO$_4$, 300 mM NaCl, 10 mM imidazole, pH 8.0) or PBS (for GST) supplemented with 10 mg ml$^{-1}$ lysozyme and EDTA-free complete protease inhibitor cocktail (Roche) using a cooled French Press cell disruptor (Thermo Scientific). Resulting lysates were incubated for 2 h at 4 °C with Ni-NTA agarose beads (Qiagen) or Glutathione Sepharose 4B beads (GE Healthcare). After incubation, beads were washed three times with Wash buffer (for 6 × His; 50 mM NaH$_2$PO$_4$, 300 mM NaCl, 20 mM imidazole, pH 8.0) or PBS (for GST). Proteins were then recovered with imidazole (for 6 × His; 50 mM NaH$_2$PO$_4$, 300 mM NaCl, 250 mM imidazole, pH 8.0) or glutathione (for GST; 10 mM Glutathione, 50 mM Tris–HCl pH 8.0) elution buffers. The eluates were dialyzed overnight at 4 °C in EDTA-free storage & concentration buffer (2 mM DTT, 100 mM KCl, 20% glycerol, 20% polyethylene glycol 8000, 20 mM HEPES pH 7.9). The amount of purified GST or 6 × His fusion proteins was verified by SDS–polyacrylamide gel electrophoresis (PAGE) stained with Coomassie technique. Five hundred nanograms of GST-tagged proteins were incubated for 2 h at 4 °C with 100 ng of His-tagged protein and 25 μl Glutathione beads in 1 ml of binding buffer (50 mM NaCl, 50 mM HEPES-KOH pH 7.6, 0.1% NP-40, 0.5% charcoal-stripped FBS, complete EDTA-free protease inhibitor cocktail). The beads were washed three times and the bound complexes were eluted in sample buffer and subjected to immunoblot analysis.

**Immunoblot analysis.** Twenty micrograms of lysates or total eluates obtained from CoIP or GST pulldown experiments were separated by SDS–PAGE and transferred electrophoretically to Immobilon-P membranes (Millipore). Membranes were blocked in 5% nonfat dry milk in PBS-Tween (0.1% Tween20 (Sigma-Aldrich)) and incubated with anti-EFTUD2 (1:10,000, Abcam; ab72456), anti-PRPF8 (1:5,000, Abcam; ab87433), anti-SNRNP200 (1:2,000, Abcam; ab118713), anti-CD2BP2 (1:2,000, Abcam; ab136141), anti-PRPF6 (1:5,000, Abcam; ab99292), anti-DDX23 (1:2,000, Abcam; ab70461), anti-SNRNP40 (1:2,000, Abcam; ab155592), anti-AAR2 (1:2,000, Abcam; ab150727), anti-RUVBL2 (1:2,000, Abcam; ab36569), anti-TSC1 (1:2,000; Thermo Scientific; 37-0400), anti-TSC2 (1:2,000, Santa Cruz; SC-893), anti-GST (1:5,000, Abcam; ab9085), anti-6 × His (1:5,000, Abcam; ab18184), anti-Myc (1:5,000, Santa Cruz; SC-40) and anti-FLAG M2 (1:5,000, Sigma-Aldrich; F1804). Protein bands were visualized using anti-rabbit and anti-mouse IgG secondary antibodies linked to horseradish peroxidase (1:2,500, GE Healthcare; NA934V and NA931V) and ECL

prime (GE Healthcare). See Supplementary Figs 3 and 4 for uncropped versions of each blot.

**Mass spectrometry and data analysis.** For non-quantitative TAP-MS and BioID-MS experiments, Desalted tryptic peptides were loaded onto a 75 μm i.d. × 150 mm Self-Pack C18 column installed in the Easy-nLC II system (Proxeon Biosystems). The buffers used for chromatography were 0.2% formic acid (buffer A) and 100% acetonitrile/0.2% formic acid (buffer B). Peptides were eluted with a two slope gradient at a flowrate of 250 nl min$^{-1}$. Solvent B first increased from 2 to 40% in 82 min and then from 40 to 80% B in 28 min. The HPLC system was coupled to a LTQ Orbitrap Velos mass spectrometer (Thermo Scientific) through a nano-ESI source (Proxeon Biosystems). Nanospray and S-lens voltages were set to 1.3–1.8 kV and 50 V, respectively. Capillary temperature was set to 225 °C. Full scan MS survey spectra ($m/z$ 1,360–2,000) in profile mode were acquired in the Orbitrap with a resolution of 60,000 with a target value at 1e6. The ten most intense peptide ions were fragmented by collision induced dissociation in the LTQ with a target value at 1e4 (normalized collision energy 35 V, activation Q 0.25, and activation time 10 ms). Target ions selected for fragmentation were dynamically excluded for 25 s. The peak list files were generated with Proteome Discoverer (version 2.1) using the following parameters: minimum mass set to 500 Da, maximum mass set to 6 kDa, no grouping of MS/MS spectra, precursor charge set to auto, and minimum number of fragment ions set to 5. Protein database searching was performed with Mascot 2.3 (Matrix Science) against the human NCBInr protein database (version July 18, 2012). The mass tolerances for precursor and fragment ions were set to 10 p.p.m. and 0.6 Da, respectively. Trypsin was used as the enzyme allowing for up to one missed cleavage. Cysteine carbamidomethylation was specified as a fixed modification, and methionine oxidation as variable modifications. In cases where multiple gene products were identified from the same peptide set, all were unambiguously removed from the data set. When multiple isoforms were identified for a unique gene, only the isoform with the best sequence coverage was reported. Proteins identified on the basis of a single spectrum were also discarded. Reliability of the data obtained from TAP-MS experiments was assessed using Decontaminator[29] which assigns a false discovery rate (FDR) to each protein–protein interactions. In the soluble fraction, the threshold FDR score was set at 0.1 although some interactions with a FDR score above 0.1 but below 0.2 were retained based on homologous proteins or components of a same protein complex having been assigned a FDR score below 0.1. Protein identification in the controls for the chromatin fraction was shown to have greater variability than that of the soluble fraction. This greatly impacted FDR assignment. For this reason, the threshold FDR score for the chromatin fraction was increased to 0.2. Probabilistic scoring was similarly performed for BioID experiments using SAINTexpress version 3.6.1 (ref. 78).

SILAC experiments were performed on the Q Exactive mass spectrometer (Thermo Scientific) but LC setup and gradient were identical to what was described for the Tap-MS experiments. LC–MS/MS data was acquired using a data-dependent top16 method combined with a dynamic exclusion window of 7 s. The mass resolution for full MS scan was set to 60,000 (at $m/z$ 1,400) and the lock mass option was enabled to improve mass accuracy. The mass range was from 360 to 2,000 $m/z$ for MS scanning with a target value at 1e6, the maximum ion fill time (IT) at 100 ms, the intensity threshold at 1.0e4 and the underfill ratio at 0.5%. The data-dependent MS2 scan events were acquired at a resolution of 17,500 with the maximum ion fill time at 50 ms and the target value at 1e5. The normalized collision energy used was at 27 and the capillary temperature was 250 °C. Quantitation was performed using the programme MaxQuant (version 1.5.1.2)[52]. Enzyme specificity was set to that of trypsin. Other parameters used were: (1) variable modifications: Methionine oxidation and Protein N Acetylation; (2) fixed modifications: Cysteine carbamidomethylation; (3) Database: UniProt human (version October 3, 2014); (4) Heavy and Medium Labels: R10K8 and R6K4; (5) MS/MS tolerance: 0.5 Da; (6) Minimum peptide length: 7; (7) Top MS/MS peaks per 100 Da: 6; (8) Maximum missed cleavages: 2; (9) Maximum of labelled amino acids: 3; (10). Proteins were considered quantified if they had at least one quantified SILAC pairs. Interactors of immunopurified EFTUD2-FLAG and PRPF8-FLAG were identified based on a $\log_2(M/L) > 1$. Interactions were then considered as significantly affected only when the $\log_2(M/H)$ ratio was greater than 1 or lower than $-1$ (twofold change or more).

**Splicing analysis.** A set of 39 units was selected from a collection of transcripts known to be alternatively spliced in HEK 293 cells. Sets of primers mapping in exons flanking the alternative splicing events were designed by using Primer3 with default parameters[79]. All forward and reverse primers were individually resuspended to 20–100 μM stock solution in Tris-EDTA buffer (IDT) and diluted as a primer pair to 1.2 μM in RNase DNase-free water (IDT). RNA integrity was assessed with an Agilent 2100 Bioanalyzer (Agilent Technologies). Reverse transcription was performed on 2.2 μg total RNA with Transcriptor reverse transcriptase, random hexamers, dNTPs and ten units of RNAse OUT (Thermo Scientific) following the manufacturer's protocol in a total volume of 20 μl. End-point PCR reactions were done on 10 ng cDNA in 10 μl final volume containing 0.2 mmol l$^{-1}$ each dNTP, 1.5 mmol l$^{-1}$ MgCl$_2$, 0.6 μmol l$^{-1}$ each primer, and 0.2 units of Platinum Taq DNA polymerase (Thermo Scientific).

An initial incubation of 2 min at 95 °C was followed by 35 cycles at 94 °C 30 s, 55 °C 30 s and 72 °C 60 s. The amplification was completed by 2 min incubation at 72 °C. PCR reactions are carried on thermocyclers GeneAmp PCR System 9700 (Thermo Scientific), and the amplified products were analysed by automated chip-based microcapillary electrophoresis on Caliper LC90 instruments (Perkin Elmer). Amplicon sizing and relative quantitation was performed by the manufacturer's software, before being uploaded to the LIMS database.

**Data availability.** The authors declare that all data supporting the findings of this study can be found within the paper and its Supplementary Information files. Protein–protein interaction data have been made public on BioGRID (https://thebiogrid.org/dataset/cloutier2017). Raw mass spectrometric data has been uploaded to the proteomics data depository PRIDE (PXD006198, PXD006199 and PXD006200) and full results of the alternative splicing assay can be found on RNOMICS PALACE (http://rnomics.med.usherbrooke.ca/palace?-purl=pcrreactiongroup/list/315). All other data are available from the corresponding author upon request.

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

## Acknowledgements

We would like to thank Drs Philippe Roux and John Blenis for sharing reagents pertaining to the TSC1–TSC2 complex and Dr Anne-Claude Gingras for providing us

with expression vectors for BioID-MS experiments. We are also grateful to the members of the IRCM Proteomics Discovery Platform (Josée Champagne, Sylvain Tessier and Marguerite Boulos) for their help in the analysis of mass spectrometric data and the members of the RNomics platform at Université de Sherbrooke (Elvy Lapointe, Roscoe Klinck and Philippe Thibault) for the design and analysis of splicing profiles. B.Ch. holds the Pierre C. Fournier Research Chair in Functional Genomics. B.Co. holds the IRCM Bell-Bombardier Research Chair. This project was financed in part by a grant of the Canadian Institutes of Health Research (CIHR; MOP-136948) awarded to B.Ch. and a grant from the Fonds de Recherche du Québec—Santé (FRQS) to B.Co.

## Author contributions

Conceptualization, P.C.; Methodology, P.C.; Formal analysis, P.C., C.P., O.H. and D.F.; Investigation, P.C., M.D., E.F.-M. and A.B.; Writing—Original draft, P.C.; Writing—Review & editing, D.F., B.Co. and B.Ch.; Funding acquisition, B.Co. and B.Ch.; Supervision, B.Co. and B.Ch.

## Additional information

**Competing interests:** The authors declare no competing financial interests.

