## [Peer Review File · Nature Communications]

Reviewers' comments:

Reviewer #1 (Remarks to the Author):

The authors carry out proteomic analysis of the interactors of the critical chaperone complex R2TP. In mammals, the complex consists of four subunits: RuvBL1, RuvBL2, RPAP3, and PIH1D1 as well as components of a prefoldin-like complex. R2TP is already known to be involved in many processes and in the assembly of several other critical complexes. Cloutier et al. mainly use pulldowns followed by mass spectrometry to identify interactors. This is a 'continuation' of an earlier study by the authors using better mass spectrometry techniques. In this study, Cloutier et al. find two additional complexes that interact with R2TP, namely TSC1-TSC2 complex (an inhibitor of mTOR) and the U5snRNP (a component of the spliceosome). As part of this study, the authors show that one function of the zinc finger HIT domains are to interact with RuvBL2.

In general, the study is well-controlled with the mass spectrometry analysis done in a very careful manner. However, the manuscript does not go beyond providing a list of interactors of R2TP. Furthermore, in many cases the authors do not verify that an interaction is indeed with R2TP as a complex rather than, for example, with a RuvBL1/RuvBL2 complex. This is especially important since RuvBL1/RuvBL2 are known to be part of many other complexes that do not include the other two subunits (RPAP3 and PIH1D1).

Specific comments:

1. Stronger evidence is needed to propose that ZNHIT2 is required to regulate the assembly of U5 snRNP. At the end of introduction, the authors state that, "disruption of ZNHIT2 and RUVBL2 expression levels affect U5 snRNP protein composition". However, this is not completely true as no clear difference was found during their siRNA experiments (Fig 7C and D). The authors suggest this is because knockdown of ZNHIT2 might not be enough to completely disrupt association between RUVBL1/2 with U5 snRNP. Perhaps a better knockout is needed. Figure 5C and D show that a ZNHIT2 mutant missing the zf-HIT domain is able to completely disrupt binding with RUVBL1 and 2. Perhaps co-expression of this mutant along with knockdown of WT ZNHIT2 can show changes in U5 snRNP composition.

2. Alternate pathways for R2TP/PFDL interaction with U5 snRNP should be explored, especially when evidence for ZNHIT2 is not completely convincing. For example, the authors mention ECD as another potential bridging factor several times in the paper, but did not explore this any further. ECD and NCDN were suggested as "novel bona fide U5 snRNP associated factors" but no co-IPs were done to confirm their mass spectrometry result. Because they were identified repeatedly in U5 snRNP purification, it is especially important to confirm this interaction.

3. The second subheading of their results state that "R2TP/PFDL is a major interactor of the TSC1-TBC1D7 subcomplex". However, no experiments were performed to confirm existence of this subcomplex. Also, no experiments were performed to confirm interaction between TBC1D7 and R2TP/PFDL either.

4. There were several interesting results that the authors raised from their mass spectrometry which did not coincide with the literature. However, the authors did not verify the validity of these interactions. Example:

Second paragraph of third subsection: AAR2 purification pulled down SNRNP200, which were thought to be mutually exclusive.

Same paragraph: CD2BP2 copurified with subunits of U4/U6.U5 tri-snRNP, U1 and U2 snRNPs, but previously thought to be absent from the tri-SNP.

5. Figure 2B – The authors should WB for all components of R2TP to confirm that the interactions are indeed with R2TP and individual subunits.

6. In their R2TP pulldowns, the authors do not seem to detect Hsp90. The R2TP-Hsp90 interaction might be weak. The authors should comment on this.

7. On page 22, the authors claim that most "genes encoding R2TP subunits are non-essential in yeast". This is not true. As I am sure the authors know that yeast RuvBL1 and RuvBL2 are essential in yeast.

Reviewer #2 (Remarks to the Author):

This manuscript describes a multi-pronged affinity purification approach to refine the model of how a co-chaperone complex regulates the assembly of the U5 snRNP. The biochemistry seems very well done and comprehensive, although TAP-MS is now a pretty old technology. It would be interesting to see if these complexes reported here co-migrate with some of the newer methods for proteomic evaluation of the interactome. Nonetheless, I cannot find anything to criticize in the biochemistry or proteomics.

Reviewer #3 (Remarks to the Author):

In this article Cloutier et al. report on the identification of new interacting complexes/proteins for R2TP/PFDL, specifically the splicesomal U5snRNP and an inhibitor complex of mTOR termed TSC1-TSC2. Using affinity purification strategies it is further shown that ZNHIT2 links the interaction of R2TP/PFDL with U5 snRNP. Importantly, the zinc finger HIT domain, present in ZNHIT2 and other ZNHIT family members, is identified as a RUVBL2-binding module. Lastly, it is shown that the lowering of ZNHIT2 and RUVBL2 expression levels affect U5snRNP integrity. Based on these data the authors suggest a role for R2TP/PFDL in assembly of U5 snRNP.

In general this is an interesting and experimentally well-performed study that uncovers new biochemical links between the R2TP chaperone system and cellular complexes. A lot of new interactions are described and provide the basis for future studies on the chaperone system. The following aspects should be addressed to substantiate the conclusions drawn by the authors and to improve the manuscript:

The manuscript is extremely difficult to read, which is mostly due to the large number of different factors that have been defined and the general complexity of the chaperone system described. As the association of TSC1/2 is only described but no additional functional data are provided, I would suggest focusing in this manuscript on the link of R2TP to the formation of U5snRNP.

The link of R2TP to U5snRNP is certainly very interesting and it would be nice if the authors could provide some mechanistic insight into this assembly event. Is there an in vitro assembly assay that could be used to directly assay this activity?

Fig 7C: top left panel: The factor ARL6IP4 is linked to alternate splicing. Since the authors relate ZNHIT2 to alternate splicing and this factor is strongly down-regulated in the U5snRNP immunoprecipitation in the absence of ZNHIT2, the authors should at least include this factor in their discussion. Experiments addressing this interesting link would also strengthen the manuscript.

Additional comments:

- The discussion needs to be focused. Parts of the results section belong to the discussion paragraph.
- The text is full of mis-spelling and grammatical errors, which should be corrected.
- Abstract: The abbreviation PFDL is not specified.
- The sentence : „A more general function for the zinc finger HIT domain in binding the RUVBL2 in

a nucleotide-dependent manner is exposed.” is not understandable without detailed knowledge of the topic-rewrite this sentence/parts of the abstract to make it understandable for a general readership.

- First sentence of Introduction: It is not clear to me what “functional morphological states” of proteins mean. Do the authors mean conformational states?

We thank the reviewers for their insightful comments and suggestions. All reviewers' comments have been addressed (see below), a number necessitating the addition of new data (see Fig. S1, Table S2, Table S5, as well as updated Fig. 1, Fig. 4, Fig. 7 and Fig. S2), thereby improving significantly the quality of our manuscript and its impact. We trust that our revised manuscript is now suitable for publication in Nature Communications.

Reviewer #1

The authors carry out proteomic analysis of the interactors of the critical chaperone complex R2TP. In mammals, the complex consists of four subunits: RuvBL1, RuvBL2, RPAP3, and PIH1D1 as well as components of a prefoldin-like complex. R2TP is already known to be involved in many processes and in the assembly of several other critical complexes. Cloutier et al. mainly use pulldowns followed by mass spectrometry to identify interactors. This is a 'continuation' of an earlier study by the authors using better mass spectrometry techniques. In this study, Cloutier et al. find two additional complexes that interact with R2TP, namely TSC1-TSC2 complex (an inhibitor of mTOR) and the U5snRNP (a component of the spliceosome). As part of this study, the authors show that one function of the zinc finger HIT domains are to interact with RuvBL2.

In general, the study is well-controlled with the mass spectrometry analysis done in a very careful manner. However, the manuscript does not go beyond providing a list of interactors of R2TP. Furthermore, in many cases the authors do not verify that an interaction is indeed with R2TP as a complex rather than, for example, with a RuvBL1/RuvBL2 complex. This is especially important since RuvBL1/RuvBL2 are known to be part of many other complexes that do not include the other two subunits (RPAP3 and PIH1D1).

Specific comments:

1. Stronger evidence is needed to propose that ZNHIT2 is required to regulate the assembly of U5 snRNP. At the end of introduction, the authors state that, "disruption of ZNHIT2 and RUVBL2 expression levels affect U5 snRNP protein composition". However, this is not completely true as no clear difference was found during their siRNA experiments (Fig 7C and D). The authors suggest this is because knockdown of ZNHIT2 might not be enough to completely disrupt association between RUVBL1/2 with U5 snRNP. Perhaps a better knockout is needed. Figure 5C and D show that a ZNHIT2 mutant missing the zf-HIT domain is able to completely disrupt binding with RUVBL1 and 2. Perhaps co-expression of this mutant along with knockdown of WT ZNHIT2 can show changes in U5 snRNP composition.

Knockdown of ZNHIT2 did affect composition of the U5 snRNP as RUVBL1 and RUVBL2 were less abundant in both PRPF8 and EFTUD2 purifications. While these proteins are not canonical subunits of the ribonucleoprotein per se, it is nonetheless an interaction of U5 snRNP that is affected by ZNHIT2 knockdown. That being said, co-transfection of the ZNHIT2 mutant lacking its zf-HIT domain with our ZNHIT2-targeting siRNAs was attempted. Unfortunately, the siRNAs used in this experiment mostly target the coding sequence of ZNHIT2 outside of its zf-HIT domain and therefore resulted in diminished expression of the mutant which likely resulted in concomitant reduced knockdown efficiency of the wild-type protein. Composition of the U5 snRNP was indeed affected, although not to the extent of direct RUVBL2 knockdown (see below).

2.

Alternate pathways for R2TP/PFDL interaction with U5 snRNP should be explored, especially when evidence for ZNHIT2 is not completely convincing. For example, the authors mention ECD as another potential bridging factor several times in the paper, but did not explore this any further. ECD and NCDN were suggested as “novel bona fide U5 snRNP associated factors” but no co-IPs were done to confirm their mass spectrometry result. Because they were identified repeatedly in U5 snRNP purification, it is especially important to confirm this interaction.

Additional quantitative SILAC-based FLAG purification were done to assess whether ECD affects integrity of the U5 snRNP. Knockdown of ECD alone and ECD in conjunction with ZNHIT2 alter composition of the U5 snRNP. In particular association of PRPF6 and CD2BP2 with PRPF8 is repeatedly decreased. Results were added to the supplementary material (see new Supp Fig. S1).

New TAP-MS assays were done for ECD and NCDN. In the case of ECD, the purification confirms interaction with U5 snRNP. Surprisingly, NCDN purification yielded a number of enzymes some of which have a role in lipid metabolism. However, interactions cannot always be confirmed by reciprocal purification experiments. There are a number of reasons why one protein can purify enough of a binding partner to allow its detection by mass spectrometry, but not the other way around (expression levels, differing specificity spectra, interference from the affinity tag, etc.). The fact remains that EAPP, TSSC4 and ZNHIT2 (but none of the other >200 different TAP-MS assays done over the years by our group) all copurified significant amounts of NCDN as did previously reported purification of murine PRPF8/Prp8 (Hutchins et al. 2010). For these reasons, we insist that the protein be considered a bona fide interactor of U5 snRNP and look forward to investigate how lipid signalling pathways might have a role in splicing regulation.

3. The second subheading of their results state that “R2TP/PFDL is a major interactor of the TSC1-TBC1D7 subcomplex”. However, no experiments were performed to confirm existence of this subcomplex. Also, no experiments were performed to confirm interaction between TBC1D7 and R2TP/PFDL either.

The subcomplex has been reported previously by Dibble et al. (2012). Very few independent investigations have provided data to substantiate this interaction which is why we felt it necessary to point out results from our own analysis (as we did for the interaction of the TSC complex and Ragulator/Rag GTPases).

Furthermore, CoIP-WB confirmations of AP-MS results is in our opinion unnecessary as western blotting is overall less reliable and inherently biased compared to mass spectrometry. However it is a common request from reviewers not actively working in proteomics and we have made it a habit to include such an assay in our manuscripts. We typically choose one or two subunits for each complexes (PRPF8 and EFTUD2 in the case of U5 snRNP and TSC1 and TSC2 in the case of the TSC complex). Exclusion of TBC1D7 was in no way, shape or form an attempt to mislead readers of this manuscript. TAP-MS was indeed done on all three subunits of the TSC complex and they do confirm interaction with R2TP/PFDL as was discussed in the manuscript.

4. There were several interesting results that the authors raised from their mass spectrometry which did not coincide with the literature. However, the authors did not verify the validity of these interactions. Example:

Second paragraph of third subsection: AAR2 purification pulled down SNRNP200, which were thought to be mutually exclusive.

Same paragraph: CD2BP2 copurified with subunits of U4/U6.U5 tri-snRNP, U1 and U2 snRNPs, but previously thought to be absent from the tri-SNP.

We agree with the reviewer that the results are interesting and thus warrant at the very least anecdotal reference but fall outside the scope of the present manuscript (novel interactors of R2TP/PFDL complex) which is why we chose not to do additional assays to confirm these results. We plan on studying further the newly-identified interactors of the U5 snRNP and these will be investigated in further details later.

5. Figure 2B – The authors should WB for all components of R2TP to confirm that the interactions are indeed with R2TP and individual subunits.

Additional antibodies were ordered to detect U5 snRNP subunits PRPF6, DDX23, SNRNP40 and AAR2. As expected, no discernable downregulation was observed for either of these proteins. The figure has been updated accordingly (Fig. 7).

6. In their R2TP pulldowns, the authors do not seem to detect Hsp90. The R2TP-Hsp90 interaction might be weak. The authors should comment on this.

Hsp90 isoforms, as well as Hsp70 isoforms, have indeed been identified in R2TP/PFDL purifications just as they had been in our previously published works. However, the increased sensitivity of the mass spectrometer as well as more stringent lysis conditions have increased the amount of protein chaperones in control samples used to calculate reliability scores. Increasing FDR cut-offs could easily salvage these interactions, but at the risk of including a number of spurious interactions. Since the main goal of this work was to identify novel interactions of R2TP/PFDL with a high level of confidence, we believe it is best to keep the highest specificity

criteria possible, even at the cost of failing to recognize true interactors that have been validated in numerous previous publications.

7. On page 22, the authors claim that most “genes encoding R2TP subunits are non-essential in yeast”. This is not true. As I am sure the authors know that yeast RuvBL1 and RuvBL2 are essential in yeast.

Indeed, RuvB1 and RuvB2 are essential gene, we meant subunits specific to R2TP, i.e. Tah1 and Pih1 (and to a certain extent Bud27, although a link between the URI1 ortholog has yet to be made with yeast R2TP). Lethality of Rvb1 and Rvb2 mutants cannot be ascribed solely to R2TP, as they are part of a number of other protein complexes. The sentence has since been clarified.

Reviewer #2:

This manuscript describes a multi-pronged affinity purification approach to refine the model of how a co-chaperone complex regulates the assembly of the U5 snRNP. The biochemistry seems very well done and comprehensive, although TAP-MS is now a pretty old technology. It would be interesting to see if these complexes reported here co-migrate with some of the newer methods for proteomic evaluation of the interactome. Nonetheless, I cannot find anything to criticize in the biochemistry or proteomics.

Purification of ZNHIT2 was undertaken using a more recently-developed technology, BioID-MS, and the association with both the U5 snRNP and R2TP/PFDL cochaperone complex are observed as reported previously by TAP-MS. The new data has been included in supplementary Table 2.

Reviewer #3:

In this article Cloutier et al. report on the identification of new interacting complexes/proteins for R2TP/PFDL, specifically the spliceosomal U5snRNP and an inhibitor complex of mTOR termed TSC1-TSC2. Using affinity purification strategies it is further shown that ZNHIT2 links the interaction of R2TP/PFDL with U5 snRNP. Importantly, the zinc finger HIT domain, present in ZNHIT2 and other ZNHIT family members, is identified as a RUVBL2-binding module. Lastly, it is shown that the lowering of ZNHIT2 and RUVBL2 expression levels affect U5snRNP integrity. Based on these data the authors suggest a role for R2TP/PFDL in assembly of U5 snRNP.

In general this is an interesting and experimentally well-performed study that uncovers new biochemical links between the R2TP chaperone system and cellular complexes. A lot of new interactions are described and provide the basis for future studies on the chaperone system. The following aspects should be addressed to substantiate the conclusions drawn by the authors and to improve the manuscript:

The manuscript is extremely difficult to read, which is mostly due to the large number of different factors that have been defined and the general complexity of the chaperone system

described. As the association of TSC1/2 is only described but no additional functional data are provided, I would suggest focusing in this manuscript on the link of R2TP to the formation of U5snRNP.

The TSC complex is a very significant new interactor that warrants being reported and elaborated on as it is new evidence of R2TP/PFDL being involved in the mTOR signalling pathway. As mentioned in the manuscript, the TSC complex was monitored for assembly following knockdown of various R2TP/PFDL subunits but no effect was observed. This should not, however, be considered as an absence of data as this could point to a regulatory role for the TSC complex in controlling R2TP/PFDL function in stress conditions. A large part of the discussion is built upon this hypothesis and removing identification of the TSC complex as a novel interactor would consequently weaken the rationale. The exact function of this interaction is still under active investigation in our laboratory.

The link of R2TP to U5snRNP is certainly very interesting and it would be nice if the authors could provide some mechanistic insight into this assembly event. Is there an in vitro assembly assay that could be used to directly assay this activity?

To our knowledge, no such assay exists as very few articles have focused on the assembly of the U5 snRNP (beyond loading of the Sm core onto the snRNA and reported role of AAR2 as an assembly factor). Furthermore, and as reported in the manuscript, the presence of subunits of the NTC complex may suggest implication in the recycling of U5 snRNP as opposed to/in addition to de novo biogenesis. Understanding the exact mechanistic of U5 snRNP biogenesis/recycling will likely take years, but it is an aspect we plan on study in the future using more focused analysis of the snRNP U5 and its interactors.

Fig 7C: top left panel: The factor ARL6IP4 is linked to alternate splicing. Since the authors relate ZNHIT2 to alternate splicing and this factor is strongly down-regulated in the U5snRNP immune-precipitation in the absence of ZNHIT2, the authors should at least include this factor in their discussion. Experiments addressing this interesting link would also strengthen the manuscript.

ARL6IP4/SFRS20 was identified with only a few spectral counts and was not observed to be modulated in label-free replicates and was thus annotated in gray. Furthermore, the protein was never identified in any of the TAP purifications of U5 snRNP or associated factors.

Additional comments:

- The discussion needs to be focused. Parts of the results section belong to the discussion paragraph.

Some of the aspects covered in the results section have been moved to the discussion. However due to the far-reaching nature of system biology research we deem it necessary to briefly point out the significance of certain results even if they do not necessarily align with the overall theme as to not distract readers from the core message elaborated in the discussion.

- The text is full of mis-spelling and grammatical errors, which should be corrected.

The manuscript has been revised carefully and misspellings have been corrected.

- Abstract: The abbreviation PFDL is not specified.

The abbreviation is now specified.

- The sentence : “A more general function for the zinc finger HIT domain in binding the RUVBL2 in a nucleotide-dependent manner is exposed” is not understandable without detailed knowledge of the topic-rewrite this sentence/parts of the abstract to make it understandable for a general readership.

The sentence has been simplified.

- First sentence of Introduction: It is not clear to me what “functional morphological states“ of proteins mean. Do the authors mean conformational states?

The sentence has been changed accordingly.

REVIEWERS' COMMENTS:

Reviewer #1 (Remarks to the Author):

The authors mainly provided arguments to address the concerns raised.

No further comments.

Reviewer #2 (Remarks to the Author):

The authors have addressed my comments

Reviewer #3 (Remarks to the Author):

The authors have addressed most points raised by the three reviewers in a satisfactory manner. Even though the relevance of some aspect of the study remain uncertain and deserve more experimental work (for example the link of R2TP to U5 assembly, association of TSC1/2 etc.), such studies would be beyond the scope of the present manuscript. I therefore think that the study in its present form provides a good basis for further studies in an active and important field of research. The work is of very high technical standard.